

# The dark cube: dark and light character profiles

Danilo Garcia[1,2,3,4,5] and Patricia Rosenberg[1,3]

[1] Blekinge Center for Competence, Blekinge County Council, Karlskrona, Sweden
[2] Department of Psychology, University of Gothenburg, Gothenburg, Sweden
[3] Network for Empowerment and Well-Being, University of Gothenburg, Lyckeby, Sweden
[4] Institute of Neuroscience and Physiology, Sahlgrenska Academy, Gothenburg, Sweden
[5] Department of Psychology, Lund University, Lund, Sweden

Corresponding authors
Danilo Garcia,
danilo.garcia@icloud.com
Patricia Rosenberg,
patriciarosenberg75@gmail.com

## ABSTRACT

**Background.** Research addressing distinctions and similarities between people's malevolent character traits (i.e., the Dark Triad: Machiavellianism, narcissism, and psychopathy) has detected inconsistent linear associations to temperament traits. Additionally, these dark traits seem to have a common core expressed as uncooperativeness. Hence, some researchers suggest that the dark traits are best represented as one global construct (i.e., the unification argument) rather than as ternary construct (i.e., the uniqueness argument). We put forward the dark cube (cf. Cloninger's character cube) comprising eight dark profiles that can be used to compare individuals who differ in one dark character trait while holding the other two constant. Our aim was to investigate in which circumstances individuals who are high in each one of the dark character traits differ in Cloninger's "light" character traits: self-directedness, cooperativeness, and self-transcendence. We also investigated if people's dark character profiles were associated to their light character profiles.

**Method.** A total of 997 participants recruited from Amazon's Mechanical Turk (MTurk) responded to the Short Dark Triad and the Short Character Inventory. Participants were allocated to eight different dark profiles and eight light profiles based on their scores in each of the traits and any possible combination of high and low scores. We used three-way interaction regression analyses and $t$-tests to investigate differences in light character traits between individuals with different dark profiles. As a second step, we compared the individuals' dark profile with her/his character profile using an exact cell-wise analysis conducted in the ROPstat software (http://www.ropstat.com).

**Results.** Individuals who expressed high levels of Machiavellianism and those who expressed high levels of psychopathy also expressed low self-directedness and low cooperativeness. Individuals with high levels of narcissism, in contrast, scored high in self-directedness. Moreover, individuals with a profile low in the dark traits were more likely to end up with a profile high in cooperativeness. The opposite was true for those individuals with a profile high in the dark traits. The rest of the cross-comparisons revealed some of the characteristics of human personality as a non-linear complex dynamic system.

**Conclusions.** Our study suggests that individuals who are high in Machiavellianism and psychopathy share a unified non-agentic and uncooperative character (i.e., irresponsible, low in self-control, unempathetic, unhelpful, untolerant), while individuals high in narcissism have a more unique character configuration expressed as high agency

and, when the other dark traits are high, highly spiritual but uncooperative. In other words, based on differences in their associations to the light side of character, the Dark Triad seems to be a dyad rather than a triad.

"There's been an awakening. Have you felt it? The Dark side, and the Light."

From the movie Star Wars: The Force Awakens.

Dark Triad Theory (*Furnham, Richards & Paulhus, 2013*) posits that people's malevolent character is represented by three dark traits: Machiavellianism, narcissism, and psychopathy (cf. *Paulhus & Williams, 2002*). Machiavellianism is expressed as a personality characterized as cold, manipulative, and with a cynical worldview and lack of morality (*Christie & Geis, 1970*). Narcissism is the tendency to lack empathy, have fantasies of enormous power, beauty and success, and at the same time have problems with criticism and show exploitativeness and exhibitionism (*Raskin & Hall, 1979*). Psychopathy is expressed as low empathy, low anxiety, and high impulsive and thrill-seeking behavior (*Hare, 1985*). Nevertheless, whether the dark traits are three distinctive traits (i.e., uniqueness hypothesis) or are one global trait (i.e., unification hypothesis) is still under debate.

In line with the uniqueness hypothesis, research suggests that individuals who score high on each one of the dark traits also display different behaviors (*Jones & Figueredo, 2013*; *Hawley, 2003*). For example, while individuals who score high on either Machiavellianism or psychopathy can be defined as manipulative, individuals who score high on Machiavellianism are more likely to use strategic planning in their manipulations, whereas individuals high on psychopathy crave quick gratification and have problems with impulse control when they manipulate others (*Brower & Price, 2001*). Additionally, individuals high on narcissism tend to manipulate others to gain self-validation with no regard to who they might hurt in doing so (*Watson et al., 1984*). In other words, although all three dark traits can be defined as manipulative at the conceptual level, the specific manipulative behavior of individuals who score high on each of the traits, is distinctive depending on which trait the individual might score high on. Accordingly, these malevolent traits are related, but the relationship is not strong, thus, the dark traits are suggested as independent from each other (*Paulhus & Williams, 2002*). Hence, this emphasizes that a person can be high on any of the traits, while being low in the others, in turn, suggesting not only differences between individuals but also within the individual.

However, factor-analytic studies show, for example, that all three of the dark traits load on the HEXACO[1] Honesty-Humility factor; suggesting that all three dark traits are negatively related to peoples' levels of sincerity, fairness, greed avoidance, and modesty (*Lee & Ashton, 2013*; see also *Furnham & Crump, 2005*). In addition, *Paulhus & Williams (2002)* found that individuals who scored high in any of the three Dark Triad traits scored low in the Big Five personality trait of agreeableness (i.e., the tendency to be kind,

[1] The HEXACO model of personality structure is a six-dimensional model of human personality based on findings from a series of lexical studies involving several European and Asian languages. The six factors, or dimensions, include Honesty-Humility (H), Emotionality (E), Extraversion (X), Agreeableness (A), Conscientiousness (C), and Openness to Experience (O). Each factor is composed of traits with characteristics indicating high and low levels of the factor. Available at https://en.wikipedia.org/wiki/HEXACO_model_of_personality_structure.

sympathetic, cooperative, warm, and considerate) and that individuals who scored high in psychopathy and narcissism also scored high on extraversion (i.e., individual's tendency to be enthusiastic, action-oriented, outgoing and enjoy interacting with people) and openness (i.e., the tendency to be open to new experiences, inquisitive, and imaginative). Individuals high in Machiavellianism and psychopathy also score low in self-discipline, lack sense of duty, and have difficulties to control, regulate, and direct their impulses (i.e., low levels of the Big Five trait of conscientiousness). These associations are in line with a unified view of the dark traits, that is, suggesting a common core (*Jakobwitz & Egan, 2006*; *Lee & Ashton, 2013*; *Paulhus & Williams, 2002*. See also *Kajonius et al., 2015*). Nevertheless, while some researchers have confirmed these results using different samples (e.g., *Lee & Ashton, 2013*), other researchers have not (e.g., *Jakobwitz & Egan, 2006*). In other words, even if there are some correlations between the Dark Triad and the Big Five, these are neither large nor consistent (*Vernon et al., 2008*). The only exception to this conclusion is the negative relationship between each of the dark traits and agreeableness (e.g., *Jakobwitz & Egan, 2006*; *Lee & Ashton, 2013*; *Paulhus & Williams, 2002*).

In sum, current research addressing distinctions in how people's dark character is related to specific personality traits, such as, extraversion, show mixed and inconsistent results. This is most likely because the personality models used to find differences or similarities between people's dark character traits only represent individuals' emotional reactions or temperament (e.g., *McAdams, 2001*; *Haidt, 2006*). After all, temperament is not useful in the distinction of who ends up with a mature or immature character (*Cloninger, 2004*). Indeed, not all individuals who are extroverts end up scoring high in psychopathy and/or narcissism (i.e., antecedent variables have different outcomes or "multi-finality") and high scores in each one of the dark traits might have different antecedents (i.e., "equifinality") (see *Cloninger & Zohar, 2011*). In other words, it might be inappropriate to assume linearity of effects in the context of personality (i.e., temperament predicts character linearly or vice versa). We argue that in order to find individual differences that can distinguish between peoples' malevolent tendencies, we need to use personality models that cover aspects of human personality that actually represent what individuals make of themselves intentionally or character.

Cloninger's personality model (*Cloninger, Svrakic & Przybeck, 1993*; *Cloninger, 2004*; *Cloninger, 2007*; *Cloninger, 2013*), for instance, comprises three "light" character dimensions: self-directedness, cooperativeness, and self-transcendence. Although Cloninger does not usually call this ternary model of character as "light," we argue that in contrast to the emotions derived from extreme expressions from our temperament traits and probably even the dark side our character (e.g., joy, sadness, disgust, fear, and anger), expressions from the character traits proposed by Cloninger are responsible for light feelings, such as, feelings of hope in that we are capable to cope with life (i.e., self-directedness), feelings of love for and from others (i.e., cooperativeness), and whether we feel connected to something bigger than ourselves (i.e., self-transcendence)—that is, a ternary unity of the being: body, mind, and psyche.[2] An individual high in self-directedness is reliable, strong, mature, goal-oriented and self-sufficient; an individual high in cooperativeness is fair, tolerant, empathetic, responsive to others needs, supportive and cooperative;

[2]Observe that the Greek word *psyche* found in *psych*ology and *psych*iatry stands for "life, soul, or spirit," which is distinct from *soma*, which refers to the "body" (*Cloninger, 2004*; see also *Cloninger & Cloninger, 2011a*; *Cloninger & Cloninger, 2011b*; *Cloninger, Salloum & Mezzich, 2012*).

# The "Light" Character Cube

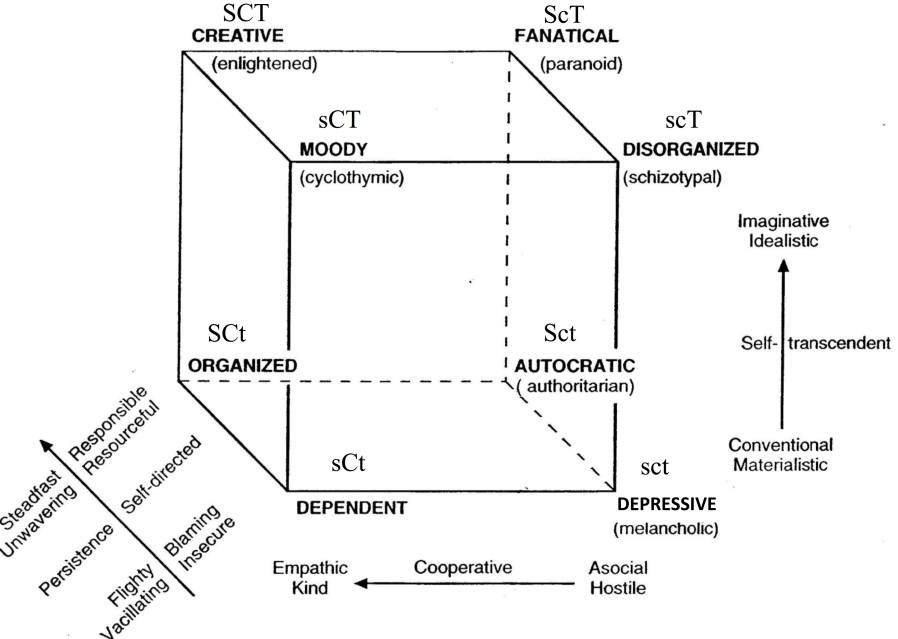

**Figure 1** **Cloninger's "light" character cube, showing all eight possible combinations of high/low scores in self-directedness, cooperativeness, and self-ranscendence.** Note: adapted with permission of Dr. C. Robert Cloninger, Center for Psychobiology of Personality, Washington University School of Medicine, St. Louis, MO. The directions of the arrows represent higher values. S, high self-directedness; s, low self-directedness; C, high cooperativeness; c, low cooperativeness; T, high self-transcendence; t, low self- transcendence.

and an individual high in self-transcendence is spiritual, satisfied, patient, selfless and creative (*Cloninger, Svrakic & Przybeck, 1993*). Cloninger has proposed that this ternary structure of character can be studied through eight different character profiles, that is, all the possible combinations of people's high and low scores in the character traits as measured by the Temperament and Character Inventory (e.g., *Cloninger & Zohar, 2011*). The creation of the character profiles using the median as the reference point allows the evaluation of the non-linear effect of each of the character traits on, for example, well-being by comparing the effect of extremes (high vs. low) of each character trait when controlling the other two. In other words, the advantage of studying multidimensional profiles of specific combinations of traits allows the understanding of the experience in an individual who is "adapting within his or her biopsychosocial context" (*Cloninger & Zohar, 2011*, p.25). See Fig. 1 for Cloninger's "light" character cube.

In the present study, we used Cloninger's character cube as an analogy to investigate if individuals' dark tendencies differ with respect to their self-concept or character (see Fig. 2). Specifically, our research question was: in which circumstances do individuals high in one dark trait express less/more self-directedness, cooperativeness and self-transcendence? We expected that, if the Dark Triad constitutes a ternary structure

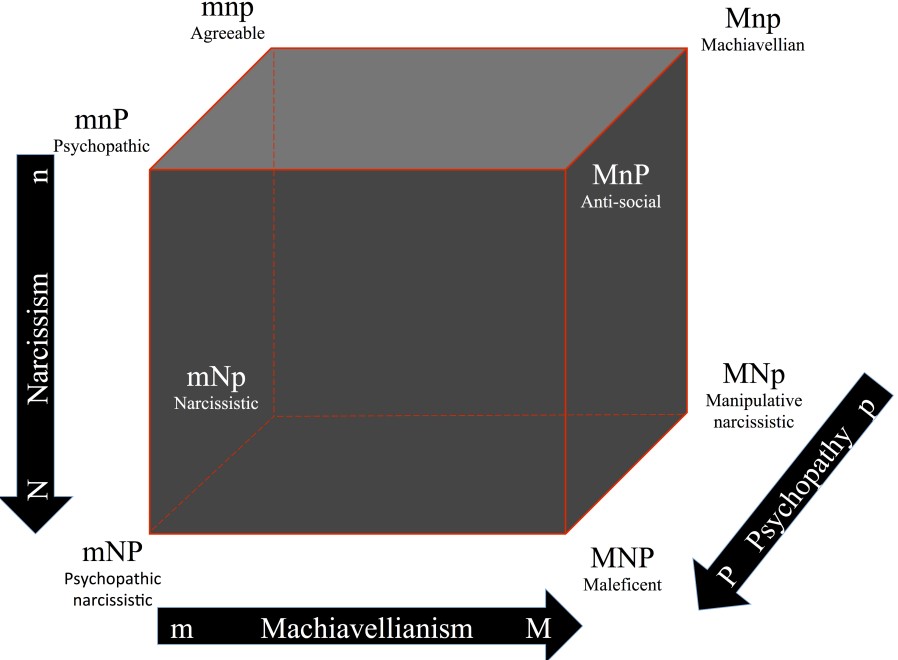

# The Dark Cube

**Figure 2** **The Dark Cube as an analogy to Cloninger's character cube, showing all eight possible combinations of high/low scores in Machiavellianism, narcissism, and psychopathy.** Note: adapted with permission from CR Cloninger. The directions of the arrows represent higher values. M, high Machiavellianism; m, low Machiavellianism; N, high narcissism; n, low narcissism; P, high psychopathy; p, low psychopathy.

of malevolent character traits, then the three dark traits should be distinguishable in the individual's goals and values in relation to oneself (i.e., self-directedness), others or society (i.e., cooperativeness), and something bigger than oneself and society, for example, the universe, nature or/and God (i.e., self-transcendence). Additionally, we also investigated if people with specific dark profiles were more likely to end up with specific character profiles.

## METHOD

### Ethical statement

After consulting with the Network for Empowerment and Well-Being's Review Board we arrived at the conclusion that the design of the present study (e.g., all participants' data were anonymous and will not be used for commercial or other non-scientific purposes) required only informed consent from the participants.

### *Participants and procedure*

Participants ($N = 1,050$) were recruited through Amazon's Mechanical Turk (MTurk; www.mturk.com/mturk/welcome) (for validation of MTurk as a data collection tool see

among others *Rand, 2011*; *Buhrmester, Kwang & Gosling, 2011*). All participants were informed that the survey was voluntary, anonymous, and that the participants could terminate the survey at any time. The MTurk workers received 50 cents (US-dollars) as compensation for participating. Two control questions were added to the survey, to control for automatic responses (e.g., "This is a control question, please answer "neither agree or disagree"). After taking away those who responded erroneously to one or both of the control questions ($n = 53$, 5.31% of all who participated), the final sample constituted 997 participants, 362 males (36.31%) and 630 females (63.19%), with an age *mean =* 34.13 years, *SD* = 12.37.

### Instruments

*The short character inventory.* This instrument was originally designed by CR Cloninger for Time Magazine as a short version of the 238-item Temperament and Character Inventory (*Cloninger, Svrakic & Przybeck, 1993*). It has never been tested empirically, but the items are all imbedded in the larger version. This makes it a brief version that is easy to administer for testing relationships among personality variables in large groups. Permission was obtained from CR Cloninger in order to include it in the present study. The inventory contains 15 items, 5 per dimension, rated on a 5-point *Likert* scale (1 = *definitely false*, 5 = *definitely true*). Examples of the items are: "Each day I try to take another step toward my goals" (self-directedness; *Cronbach's α* = .56), "I enjoy getting revenge on people who hurt me" (cooperativeness, reversed item, *Cronbach's α* = .54), and "Sometimes I have felt like I was part of something with no limits or boundaries in time and space" (self-transcendence, *Cronbach's α* = .57).

*The short dark triad inventory (*Jones & Paulhus, 2014*).* This instrument comprises 27 items, 9 per each dark trait. Examples of the items are: "Most people can be manipulated" (Machiavellianism, *Cronbach's α* = .76), "People see me as a natural leader" (narcissism; *Cronbach's α* = .74), and "Payback needs to be quick and nasty" (psychopathy; *Cronbach's α* = .73). The items were rated on a 5-point *Likert* scale (1 = *strongly disagree*, 5 = *strongly agree*).

### Statistical procedure

The sample was divided into subjects above (high) and below (low) the median for each of the three dark traits: Machiavellianism (*median =* 3.00; M for high, m for low), narcissism (*median =* 2.67; N for high, n for low), and psychopathy (*median =* 1.78; P for high, p for low). Then the participants were grouped according to all the possible combinations of high and low dark trait scores to define the eight possible Dark Triad profiles: MNP "maleficent" ($n = 232$, 23.3%), MNp "manipulative narcissistic" ($n = 66$, 6.6%), MnP "anti-social" ($n = 134$, 13.4%), Mnp "Machiavellian" ($n = 92$, 9.2%), mNP "psychopathic narcissistic" ($n = 86$, 8.6%), mNp "narcissistic" ($n = 93$, 9.3%), mnP "psychopathic" ($n = 76$, 7.6%), and mnp "agreeable" ($n = 218$, 22.0%). See Fig. 3.

We followed the same procedure using participants' "light" character scores. The sample was divided into subjects above (high) and below (low) the median for each of the three character traits: self-directedness (*median =* 3.60; S for high, s for low),

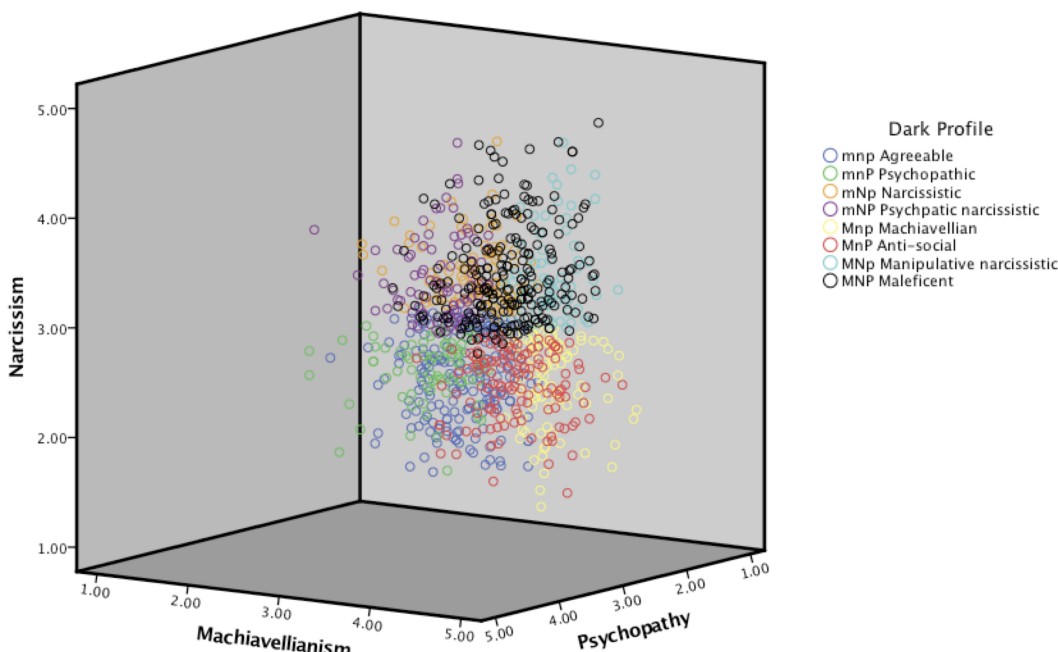

**Figure 3 Distribution of the different dark character profiles in the Dark Cube.** Note: adapted with permission from CR Cloninger. The directions of the arrows represent higher values. M, high Machiavellianism; m, low Machiavellianism; N, high narcissism; n, low narcissism; P, high psychopathy; p, low psychopathy. Observe that the scale for neuroticism does not match that of Fig. 2 due to technical issues.

cooperativeness (*median* = 3.80; C for high, c for low), and self-transcendence (*median* = 3.00; T for high, t for low). Then the participants were grouped according to all the possible combinations of high and low character trait scores to define the eight possible light character profiles: SCT "creative" ($n = 149$, 14.9%), SCt "organized" ($n = 144$, 14.4%), ScT "fanatical" ($n = 73$, 7.3%), Sct "autocratic" ($n = 94$, 9.4%), sCT "moody" ($n = 103$, 10.3%), sCt "dependent" ($n = 98$, 9.8%), scT "disorganized" ($n = 137$, 13.7%), and sct "depressive" ($n = 199$, 20.0%).

## RESULTS AND DISCUSSION

Table 1 shows the correlations, means, standard deviation, and *Cronbach's alpha* for each of the character traits. The significant correlations between the light character traits was relatively low and varied between $r = .10$ (self-directedness and self-transcendence) and .29 (self-directedness and cooperativeness), while the significant correlations between the dark character traits varied between $r = .35$ (Machiavellianism and narcissism) and .51 (Machiavellianism and psychopathy). The significant correlations between dark and light character traits varied between $r = .14$ (self-directedness and narcissism) and $-.58$ (cooperativeness and psychopathy).

Despite the fact that earlier studies using light character profiles (e.g., *Cloninger & Zohar, 2011*) have used *t-tests* to address the question of in which circumstances do individuals high in one trait express less/more of an outcome variable, we opted to first conduct three three-way interaction regression analyses using the whole scale range of

Peer**J**

**Table 1** The correlation matrix including means, standard deviations (SD) and Alphas of both light and dark character traits.

| | Character trait | Mean | SD | Alpha | Self-directedness | Cooperativeness | Self-transcendence | Machiavellianism | Narcissism | Psychopathy |
|---|---|---|---|---|---|---|---|---|---|---|
| Light | Self-directedness | 3.59 | ±0.68 | .56 | – | | | | | |
| | Cooperativeness | 3.81 | ±0.64 | .53 | .29*** | – | | | | |
| | Self-transcendence | 2.97 | ±0.77 | .56 | .10** | .16** | – | | | |
| Dark | Machiavellianism | 2.98 | ±0.67 | .74 | −.28*** | −.57*** | −.05 | – | | |
| | Narcissism | 2.73 | ±0.65 | .72 | .14*** | −.25*** | .19*** | .35*** | – | |
| | Psychopathy | 1.86 | ±0.6 | .72 | −.27*** | −.58*** | .02 | .51*** | .40*** | – |

**Notes.**

$^*p < .05$.
$^{**}p < .01$.
$^{***}p < .001$.

Yellow fields, significant correlations between light character traits.
Black fields, significant correlations between dark character traits.
Grey fields, significant correlations between light and dark character traits.
**Table 2 Three-way interaction regression analyses.** Analyses using the Dark Triad traits (i.e., Machiavellianism, narcissism, and psychopathy) as the independent variables and the Light Triad traits (i.e., self-directedness, cooperativeness and self-transcendence) as the outcome.

| Variable | Self-directedness | | Cooperativeness | | Self-transcendence | |
|---|---|---|---|---|---|---|
| | $B$ | $\Delta R^2$ | $B$ | $\Delta R^2$ | $B$ | $\Delta R^2$ |
| Step 1 | | .05 | | .44 | | .19 |
| Machiavellianism (M) | −.13** | | −.38*** | | −.27*** | |
| Narcissism (N) | .24*** | | .06** | | .35*** | |
| Psychopathy (P) | −.01 | | −.42*** | | −.27*** | |
| Step 2 | | .00 | | .00 | | .00 |
| Machiavellianism | −.13** | | −.38*** | | −.26*** | |
| Narcissism | .24*** | | .05 | | .34*** | |
| Psychopathy | .01 | | −.42*** | | −.27*** | |
| Interaction M*N | .04 | | .01 | | .05 | |
| Interaction M*P | −.05 | | −.05** | | −.01 | |
| Interaction N*P | −.01 | | .06** | | −.01 | |
| Step 3 | | .00 | | .00 | | .00 |
| Machiavellianism | −.14*** | | −.39*** | | −.26*** | |
| Narcissism | .22*** | | .04 | | .35*** | |
| Psychopathy | .01 | | −.42*** | | −.27*** | |
| Interaction M*N | .04 | | .02 | | .05 | |
| Interaction M*P | −.06 | | −.06** | | .00 | |
| Interaction N*P | −.04 | | .05 | | −.00 | |
| Interaction M*N*P | .03 | | .01 | | −.01 | |
| Total | | .05 | | .44 | | .19 |

**Notes.**
*$p = .01$.
**$p < .05$.
***$p = .000$.

the dark character traits as the predictor variables and the light character traits as the outcome. This strategy, although linear, allowed us to have higher statistical power. All variables were standardized before performing the analysis. The first model explained 19.30% of the variance in self-directedness, the second model explained 44.30% of the variance in cooperativeness, and the third model explained only 4.60% of the variance in self-transcendence.

In step 1, Machiavellianism was negatively associated to all three light character traits while narcissism was positively associated to all three light character traits. Psychopathy was negatively associated to cooperativeness and self-transcendence. In step 2, there was a small but significant negative interaction effect between Machiavellianism and psychopathy on cooperativeness and a small but significant positive interaction effect between narcissism and psychopathy on cooperativeness. In step 3, however, there was no significant three-way interaction effect between the dark character traits in relation to any of the light character traits. See Table 2 for the details. Nevertheless, seeing that we based

**Table 3** Results from the *t*-tests analyses for each Dark Triad character trait for self-directedness, cooperativeness, and self-transcendence. Significant results are marked in bold type.

| | Self-directedness | | | | Cooperativeness | | | | Self-transcendence | | | |
|---|---|---|---|---|---|---|---|---|---|---|---|---|
| | *t* | *p* | *Cohen's d* | $r_{pb}$ | *t* | *p* | *Cohen's d* | $r_{pb}$ | *t* | *p* | *Cohen's d* | $r_{pb}$ |
| **Machiavellianism** | | | | | | | | | | | | |
| MNP vs. mNP | −3.56 | .000 | −0.53 | .26 | −7.60 | .000 | −0.86 | .39 | −1.91 | .057 | −0.21 | .11 |
| MNp vs. mNp | −2.01 | .046 | −0.32 | .16 | −2.70 | .008 | −0.51 | .25 | 0.86 | .391 | 0.14 | .07 |
| MnP vs. mnP | −3.05 | .003 | −0.42 | .21 | −6.23 | .000 | −0.91 | .41 | −2.87 | .004 | −0.40 | .20 |
| Mnp vs. mnp | −3.13 | .002 | −0.36 | .18 | −6.05 | .000 | −1.01 | .45 | 0.24 | .813 | 0.03 | .01 |
| **Narcissism** | | | | | | | | | | | | |
| MNP vs. MnP | 3.56 | .000 | 0.37 | .18 | −0.84 | .401 | −0.09 | .04 | 3.73 | .000 | 0.39 | .19 |
| MNp vs. Mnp | 2.60 | .010 | 0.42 | .20 | 0.56 | .575 | 0.09 | .04 | 1.87 | .063 | 0.30 | .15 |
| mNP vs. mnP | 2.73 | .007 | 0.43 | .21 | 0.25 | .803 | 0.04 | .02 | 1.42 | .158 | 0.22 | .11 |
| mNp vs. mnp | 2.79 | .006 | 0.39 | .19 | −2.38 | .018 | −0.27 | .13 | 1.43 | .153 | 0.16 | .08 |
| **Psychopathy** | | | | | | | | | | | | |
| MNP vs. MNp | −3.23 | .001 | −0.38 | .18 | −7.86 | .000 | −0.91 | .42 | −1.42 | .157 | −0.17 | .08 |
| MnP vs. Mnp | −2.94 | .004 | −0.39 | .19 | −6.71 | .000 | −0.90 | .41 | −1.98 | .049 | −0.26 | .13 |
| mNP vs. mNp | −2.66 | .009 | −0.40 | .20 | −4.05 | .000 | −0.66 | .31 | 1.20 | .233 | 0.18 | .09 |
| mnP vs. mnp | −2.96 | .003 | −0.35 | .17 | −7.32 | .000 | −0.86 | .39 | 1.20 | .232 | 0.19 | .10 |

**Notes.**

$r_{pb}$, point-biseral coefficient.

our investigation on dark profiles analogical to Cloninger's light character profiles, rather than linear analyses we continued with comparison analyses between dark profiles.

Paired *t*-tests were performed to evaluate the non-linear influence of each of the Dark Triad profiles on the character traits. The comparisons investigated the effect of extremes of each Dark Triad profile when the other two were held constant (see Table 3 for the details). Individuals high in Machiavellianism scored, in all cases, lower in self-directedness and cooperativeness than individuals low in Machiavellianism. Thus, suggesting a clear and unique association between Machiavellian tendencies and low levels of self-acceptance, sense of autonomy and responsibility, self-control, and self-actualization (i.e., low self-directedness), and also low tolerance towards others, unhelpfulness, and low levels of empathy (i.e., low cooperativeness). Individuals high in Machiavellianism, compared to those low in Machiavellianism, scored lower in self-transcendence only when they were low in narcissism and high in psychopathy (MnP vs. mnP).

Individuals high in narcissism were, in all cases, higher in self-directedness than individuals low in narcissism. That is, in contrast to its negative relation to both Machiavellianism and psychopathy, self-directedness is positively associated to narcissism. This suggests that narcissism is distinctive from the other two traits when it comes to agentic or self-directed behavior. Moreover, individuals high in narcissism, compared to those low in narcissism, reported lower cooperativeness only when they were low in both Machiavellianism and psychopathy (mNp vs. mnp). This suggests that narcissism in its pure form is associated with unhelpfulness, low tolerance towards others, and low empathy. Finally, individuals high in narcissism, compared to those low in narcissism,

reported higher self-transcendence only when they were high in both Machiavellianism and psychopathy (MNP vs. MnP). This suggests that an individual who is high in the three dark traits might be goal-directed and unempathic (i.e., high self-directedness and low cooperativeness). At the same time she/he might have a sense of being in connection with something divine or universal (i.e., high self-transcendence). In other words, the "maleficent" dark profile (i.e., MNP) seems to correspond to a "fanatical" light character profile (i.e., ScT)—individuals with a "fanatical" profile can be characterized as independent and paranoid, and being projective of blame (*Cloninger, Bayon & Svrakic, 1998*). This is actually in accordance to descriptions of successful dictators such as Gadhafi and Saddam or terrorists such as Osama Bin Laden and Anders Breivik who have been suggested to have a "maleficent" dark character profile (i.e., MNP; see *Furnham, Richards & Paulhus, 2013*).

Individuals high in psychopathy were, in all cases, lower in self-directedness and cooperativeness when compared to individuals low in psychopathy. Thus, as for Machiavellianism, this suggests a clear and unique association between psychopathy and low self-directed behavior and uncooperativeness. Individuals high in psychopathy, compared to those low in psychopathy, reported lower self-transcendence only when they were high in Machiavellianism and low in narcissism (MnP vs. Mnp).

In the second analysis, we compared the individual's dark profile with her/his character profile using an exact cell-wise analysis in the ROPstat software (*Vargha, Torma & Bergman, 2015*, http://www.ropstat.com). The aim with this base model was to create a reference (i.e., an estimated expected cell frequency) to which we compared the observed cell frequency (see *Von Eye, Bogat & Rhodes, 2006*). In short, if a specific cell contains more cases than expected under this base model, this cell indicates a relationship that exists only in this particular sector of the cross-classification, that is, it constitutes a *type*. If a cell, in contrast contains fewer cases than expected under the base model, this cell also indicates a local relationship, that is, it constitutes an *antitype* (see also *Bergman & El-Khouri, 1987*). We examined the idea of dark profile having an effect on character profile membership (see Table 4).

For individuals with an "agreeable" dark profile (i.e., mnp: low in all dark traits) there was a higher probability (i.e., type) to end up with a high cooperative profile (i.e., SCT, SCt, sCT, sCt) and a lower probability (i.e., anti-type) of ending up with a low cooperative profile (i.e., ScT, Sct, scT, Sct). As it could be expected, the opposite was found for the "maleficent" dark profile (i.e., MNP: high in all dark traits). This is in line with research linking Machiavellian tendencies and dark tendencies per se to low agreeableness and unhelpful behavior (i.e., the unification argument). For the rest of the profiles, our analyses show a complex pattern of possible combinations. For example, although individuals with an "anti-social" dark profile (i.e., MnP) were less likely (i.e., anti-type) to end up with a high cooperative character profile, they were more likely (i.e., type) to end up with either a "disorganized" (i.e., scT) or a "depressive" character profile (i.e., sct). In other words, all individuals with an "anti-social" dark profile are uncooperative, while some of them are at the same time either highly spiritual or with underdeveloped character. Additionally, as observed in the *t*-tests analyses, the "fanatical" light character profile

Peerj

**Table 4  Exact cell-wise analysis of two-way frequencies: dark and light character profiles.**

| | CHARACTER PROFILE | | | | | | | |
|---|---|---|---|---|---|---|---|---|
| **DARK TRIAD PROFILE** | **SCT** "Creative" | **SCt** "Organized" | **ScT** "Fanatical" | **Sct** "Autocratic" | **sCT** "Moody" | **sCt** "Dependent" | **scT** "Disorganized" | **sct** "Depressive" |
| **mnp "Agreeable"** | TYPE | TYPE | ANTI-TYPE | ANTI-TYPE | TYPE | TYPE | ANTI-TYPE | ANTI-TYPE |
| Observed frequency | 51 | 53 | 5 | 10 | 33 | 42 | 6 | 18 |
| Expected frequency | 32.6 | 31.5 | 16 | 20.6 | 22.5 | 21.4 | 30 | 43.5 |
| *Chi-square* | 10.41*** | 14.70*** | 7.53*** | 5.42*** | 4.88*** | 19.75*** | 19.16*** | 14.96*** |
| **mnP "Psychopathic"** | – | – | – | ANTI-TYPE | – | – | – | – |
| Observed frequency | 8 | 11 | 7 | 2 | 9 | 7 | 11 | 21 |
| Expected frequency | 11.4 | 11 | 5.6 | 7.2 | 7.9 | 7.5 | 10.4 | 15.2 |
| *Chi-square* | 0.99 | 0 | 0.37 | 3.72* | 0.17 | 0.03 | 0.03 | 2.24 |
| **mNp "Narcissistic"** | TYPE | TYPE | – | – | – | – | ANTI-TYPE | ANTI-TYPE |
| Observed frequency | 32 | 23 | 2 | 9 | 10 | 9 | 3 | 5 |
| Expected frequency | 13.9 | 13.4 | 6.8 | 8.8 | 9.6 | 9.1 | 12.8 | 18.6 |
| *Chi-square* | 23.57*** | 6.81*** | 3.4 | 0.01 | 0.02 | 0 | 7.48*** | 9.91*** |
| **mNP "Psychopathic narcissistic"** | – | – | – | – | – | ANTI-TYPE | – | ANTI-TYPE |
| Observed frequency | 17 | 15 | 10 | 7 | 14 | 3 | 10 | 10 |
| Expected frequency | 12.9 | 12.4 | 6.3 | 8.1 | 8.9 | 8.5 | 11.8 | 17.2 |
| *Chi-square* | 1.34 | 0.54 | 2.18 | 0.15 | 2.95 | 3.52* | 0.28 | 2.99* |
| **Mnp "Machiavellian"** | – | – | – | – | – | – | – | – |
| Observed frequency | 9 | 14 | 7 | 8 | 14 | 14 | 9 | 17 |
| Expected frequency | 13.7 | 13.3 | 6.7 | 8.7 | 9.5 | 9 | 12.6 | 18.4 |
| *Chi-square* | 1.64 | 0.04 | 0.01 | 0.05 | 2.13 | 2.72 | 1.05 | 0.1 |
| **MnP "Anti-social"** | ANTI-TYPE | ANTI-TYPE | – | – | ANTI-TYPE | – | TYPE | TYPE |
| Observed frequency | 5 | 7 | 8 | 15 | 6 | 9 | 28 | 56 |
| Expected frequency | 20 | 19.4 | 9.8 | 12.6 | 13.8 | 13.2 | 18.4 | 26.7 |
| *Chi-square* | 11.27*** | 7.89*** | 0.33 | 0.44 | 4.44* | 1.32 | 4.99* | 32.00*** |

**Table 4** (*continued*)

| DARK TRIAD PROFILE | CHARACTER PROFILE | | | | | | | |
|---|---|---|---|---|---|---|---|---|
| | SCT "Creative" | SCt "Organized" | ScT "Fanatical" | Sct "Autocratic" | sCT "Moody" | sCt "Dependent" | scT "Disorganized" | sct "Depressive" |
| **MNp "Manipulative narcissistic"** | TYPE | – | – | – | – | – | – | – |
| Observed frequency | 16 | 11 | 3 | 9 | 6 | 5 | 8 | 8 |
| Expected frequency | 9.9 | 9.5 | 4.8 | 6.2 | 6.8 | 6.5 | 9.1 | 13.2 |
| *Chi-square* | 3.82[*] | 0.23 | 0.69 | 1.24 | 0.1 | 0.34 | 0.13 | 2.03 |
| **MNP "Maleficent"** | ANTI-TYPE | ANTI-TYPE | TYPE | TYPE | ANTI-TYPE | ANTI-TYPE | TYPE | TYPE |
| Observed frequency | 11 | 10 | 31 | 34 | 11 | 9 | 62 | 64 |
| Expected frequency | 34.7 | 33.5 | 17 | 21.9 | 24 | 22.8 | 31.9 | 46.3 |
| *Chi-square* | 16.16[***] | 16.49[***] | 11.56[***] | 6.72[***] | 7.02[***] | 8.36[***] | 28.46[***] | 6.76[***] |

**Notes.**

[***] $p < .001$.

[*] $p < .05$.

TYPE, the observed cell frequency is significantly greater than the expected (grey fields).

ANTI-TYPE, the observed cell frequency is significantly smaller than the expected (black fields).

–, the observed cell frequency is as expected (white fields).

(i.e., ScT) was common (i.e., type) for those who had a "maleficent" dark character profile (i.e., MNP) and less common (i.e., anti-type) for those who had an "agreeable" dark profile (i.e., mnp). See Table 4.

## Limitations

The data is self-reported and therefore subject to personal perceptual bias. Moreover, the light character scales showed low reliabilities, probably because the instrument has so few items. But the items are all imbedded in the larger version, which has been validated across many studies. This makes it a brief version that is easy to administer for testing relationships among personality variables in large groups of subjects but probably not for precise assessment of individuals. Nevertheless, given the observation of the low *Cronbach's alpha* scores, it seems appropriate that the factorial validity should be examined in future studies. In that case, researchers should consult suggestions regarding the evaluation of short questionnaires (e.g., *Marsh, Martin & Jackson, 2010*; *Olaru, Witthöft & Wilhelm, 2015*). That being said, this short version actually discerned the expected patterns among the Dark Triad profiles. The question of causality is, however, beyond the present cross-sectional study.

Some aspects related to the use of MTurk as a data collection method might influence the validity of the results, such as, workers' attention levels, cross-talk between participants, and the fact that participants get remuneration for their answers (*Buhrmester, Kwang & Gosling, 2011*). Nevertheless, a large quantity of studies show that data on psychological measures collected through MTurk meets academic standards, is demographically diverse, and also that health measures show satisfactory internal as well as test–retest reliability (*Buhrmester, Kwang & Gosling, 2011*; *Horton, Rand & Zeckhauser, 2011*; *Shapiro, Chandler & Mueller, 2013*; *Paolacci, Chandler & Ipeirotis, 2010*). In addition, the amount of payment does not seem to affect data quality; remuneration is usually small, and workers report being intrinsically motivated (e.g., participate for enjoyment) (*Buhrmester, Kwang & Gosling, 2011*).

Finally, it is plausible to argue that dichotomizing into groups that are classified as being low or high on traits will likely lead to a loss of power effectively equivalent to a loss of sample size (e.g., *MacCallum et al., 2002*). Additionally, since median splits distort the meaning of high and low, it is plausible to criticize the validity of this approach to create the profiles—scores just-above and just-below the median become high and low by arbitrariness, not by reality (*Schütz, Archer & Garcia, 2013*; *Garcia, MacDonald & Archer, 2015*). In light of these arguments, we included as a first step the three three-way interaction regression analyses, which actually confirmed the core findings. However, others have argued that from a person-centered framework personality dimensions within the individual can be seen as interwoven components with whole-system properties (cf. *Bergman & Wångby, 2014*). The outlook of the individual as a whole-system unit is then best studied by analyzing patterns of information or profiles (*Bergman & Wångby, 2014*). Although at a theoretical level there is a myriad of probable patterns of combinations of peoples' levels of personality traits, if viewed at a global level, there should be a small number of more frequently observed patterns or "common types" (*Bergman & Wångby,*

[3]Fixed points, or steady states of a given dynamical system; these are values of the variable that don't change over time. Some of these fixed points are attractive, meaning that if the system starts out in a nearby state, it converges towards the fixed point (https://en.wikipedia.org/wiki/Dynamical_systems_theory).

2014; *Bergman & Magnusson, 1997*; see also *Cloninger, Svrakic & Svrakic, 1997*, who explain nonlinear dynamics in complex adaptive systems). It is beyond the scope of the present study to discern the best possible way to arrive to the dark profiles but researchers should definitely consider putting the parts together. After all, although all possible variations of personality profiles is large, human personality is a non-linear dynamic system (*Cloninger, 2004*) that responds to the laws of attractor states,[3] which are essential for the understanding of most physical and human phenomena (*Hiver, 2014*; see also *Prigogine & Stengers, 1984*). Here we based our choice of using median splits in two presuppositions: (1) that at the conceptual and theoretical level the dark character traits constitute a ternary structure of human dark/maladaptive character and (2) that an opposite ternary structure of human light/adaptive character generates eight possible profile combinations that have been largely studied using median splits. The limitation of our approach might actually reside in the fact that the Dark Triad is not a ternary structure as the one represented by the light character traits in Cloninger's biopsychosociopiritual model of personality. Indeed, the comparison of light character profiles with the dark ones shows that the dark triad inadequately captures the anti-types already measured well by the light triad.

## CONCLUSIONS AND FINAL REMARKS

Far from the mixed patterns using the Big Five traits (e.g., *Jakobwitz & Egan, 2006*; *Paulhus & Williams, 2002*), this study suggest that Machiavellianism and psychopathy share a unified but unique non-agentic (low self-directedness) and uncooperative (low cooperativeness) character; while narcissism has a unique character configuration expressed as high agency. In other words, the Dark Triad does not seem to represent a ternary structure as Cloninger's character model, it rather is a dyad of malevolent character traits in relation to the self (i.e., narcissism) and others (i.e., both Machiavellianism and Psychopathy). Specifically, the Dark Triad lacks a dark trait that corresponds uniquely to a spiritual dimension of human character. Recently, however, *Paulhus (2014)* has suggested everyday sadism as a fourth component, making the triad into a tetrad. It is plausible that future studies might find that enjoyment of cruelty against other human beings and animals is uniquely associated to the inability of transcend the self and feel part of the whole universe.

"One of the most highly developed skills in contemporary Western civilization is dissection: the split-up of problems into their smallest possible components. We are good at it. So good, we often forget to put the pieces back together again.

This skill is perhaps most finely honed in science. There we not only routinely break problems down into bite-sized chunks and mini-chunks, we then very often isolate each one from its environment by means of a useful trick. We say *ceteris paribus*—all other things being equal. In this way we can ignore the complex interactions between our problem and the rest of the universe."

Alvin Toffler in the Foreword to Order out of Chaos: Man's New Dialogue with Nature by *Prigogine & Stengers (1984)*.

### Funding

The development of this article was funded by a grant from the Swedish Research Council (Dnr. 2015-01229). The funders had no role in study design, data collection and analysis, decision to publish, or preparation of the manuscript.

### Grant Disclosures

The following grant information was disclosed by the authors:
Swedish Research Council: 2015-01229.

### Competing Interests

Danilo Garcia is the Director of the Blekinge Center of Competence, which is the Blekinge County Council's research and development unit. The Center works on innovations in public health and practice through interdisciplinary scientific research, community projects, and the dissemination of knowledge in order to increase the quality of life of the habitants of the county of Blekinge, Sweden.

### Author Contributions

- Danilo Garcia and Patricia Rosenberg conceived and designed the experiments, performed the experiments, analyzed the data, contributed reagents/materials/analysis tools, wrote the paper, prepared figures and/or tables, reviewed drafts of the paper.

### Human Ethics

The following information was supplied relating to ethical approvals (i.e., approving body and any reference numbers):

After consulting with the Network for Empowerment and Well-Being's Review Board we arrived at the conclusion that the design of the present study (e.g., all participants' data were anonymous and will not be used for commercial or other non-scientific purposes) required only informed consent from the participants.

### Data Availability

The raw data is available upon request to the Network for Empowerment and Well-Being, lead researcher Danilo Garcia: http://celam.gu.se/people/danilo-garcia or http://ltblekinge.se/Forskning-och-utveckling/Blekinge-kompetenscentrum/Summary-in-English/.

### Supplemental Information

Supplemental information for this article can be found online at http://dx.doi.org/10.7717/peerj.1675#supplemental-information.

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
