# Peer review of "The dark cube: dark and light character profiles"

_PeerJ, doi:10.7717/peerj.1675_

## Round 0.1 · original submission · Major Revisions

Two reviewers with expertise in areas that are related to the focus of your manuscript have examined your submission and provided critiques for your consideration; their comments are included at the bottom of this letter. I too have reviewed your submission, but primarily from a methodological standpoint since I do not have expertise on the substantive focus of the study. I have two major comments to offer in this regard, one of which echoes and extends a major comment from Reviewer 1.

1. Given that the Short Character Inventory has not yet been tested empirically, and the observation of the low Cronbach alpha scores, it seems appropriate that the factorial validity of the tool is examined in the present sample prior to conducting the main analyses. You might like to consult the work of Herb Marsh and his colleagues with regard to the evaluation of short-forms of questionnaires.

Marsh, H.W., Martin, A.J., & Jackson, S. (2010). Introducing a short version of the Physical Self Description Questionnaire: New strategies, short-form evaluative criteria, and applications of factor analyses. Journal of Sport and Exercise Psychology, 32, 438-482.

2. I agree with Reviewer 1 that artificially creating groups based on a median split does not appear warranted. Dichotomizing into groups that are classified as being low or high on dark traits, for example, will likely lead to a loss of power effectively equivalent to a loss of sample size (doi: 10.1037//1082-989X.7.1.19). Alternative data analytic approaches include 3 x linear regressions with the different character dimensions as the dependent variables as suggested by Reviewer 1, as well as using structural equation modeling (SEM) to simultaneously model all 3 dependent variables and the various interaction terms. Statistical software such as Mplus permits analysts to create latent variable interactions within SEM (see https://www.statmodel.com/download/LV%20Interaction.pdf OR doi: 10.1177/0165025414552301).

As these points can be addressed in a revised submission, I am happy to offer you the opportunity to revise and resubmit the manuscript.

Sincerely,

Daniel Gucciardi, PhD

Reviewer 1 ·

Basic reporting

Please see general comments.

Experimental design

Please see general comments.

Validity of the findings

Please see general comments.

Additional comments

This is a nicely written and neatly prepared paper. However, I question the appropriateness of the statistical analyses applied.

I do not think that median splits and t-tests are the best analytic strategy. It is widely known that artificially creating groups with median splits result in low statistical power and in the present manuscript a large number of t-tests are reported. Of course, sample size was large here, but I wonder if linear regression analyses are a better option. For example, three linear regression analyses (for self-directedness, cooperativeness, and self-transcendence as outcomes) could be calculated with Machiavellianism, Psychopathy, and Narcissism scores as predictor variables, and also all two-way interactions and the three-way interaction M x P x N. If there are significant interactions, these can be followed up at high and low values as described by Aiken, L. S., & West, S. G. (1991). Multiple regression: Testing and interpreting interactions. Thousand Oaks, CA: Sage.
Such analyses can be easily calculated using PROCESS for SPSS, which is freely available here: http://www.processmacro.org/
The reference is Hayes, A. F. (2013). Introduction to Mediation, Moderation, and Conditional Process Analysis. New York: The Guilford Press.
This strategy has three important advantages over the current analytic strategy: Higher statistical power, easy inclusion of important covariates (here: gender; this can be included as a single predictor or even with interactive effects, i.e. gender x P, gender x M, gender x N, etc.) and a substantial decrease in the number of calculations. For self-directedness, for example, I would expect that scores on all three subscales are significant predictors (N positively associated with SD and P/M negatively related to SD) with no significant interaction effects. Thus, this single calculation includes the exact same information as the 12 comparisons that are currently reported in the first column in table 1.

I am not familiar with ROPstat. However, observed frequencies in many cells are smaller than 10 or even 5 and I assume that this is inappropriate for any kind of such analyses. Although I am not an expert in these analyses either, I think that using all questionnaire scores as continuous measures in cluster analysis or latent profile analysis would be a better option than using these groups based on median splits.


Minor points:

Line 149: I think this sentence should read „we investigated … were more likely…”.

Line 167: Please add percentage values to the number of males and females. Please add a unit for mean age (years).

Lines 188-190 & 196-198: Please state descriptive statistics here (i.e., medians).

Line 194: typo

Statistical analyses:
Descriptive statistics (means and SDs) should be reported. Maybe table 1 can be split into three tables for the sake of clarity.

Figure 1:
Please include a statement in the figure caption explaining that lower case letters represent low scores and upper case letters represent high scores on the respective subscale and that the direction of the arrows represent higher values as well.

·

Basic reporting

The writing is clear and concise. The introduction and discussion are informative and balanced.

Experimental design

This is an association study in a large sample (n = 997) using two brief personality inventories for "dark"/maladaptive versus "light"/adaptive aspects of character. Multidimensional profiles allow detection of interactions among each set of traits and for tests of covariance across the inventories. Interpretation is primarily limited to the number of independent dimensions represented in the two inventories.

Validity of the findings

The authors find that the dark triad is well-associated with Self-directedness and Cooperativeness in the short character inventory but does not have significant associations with Self-transcendence. They properly conclude that the Dark Triad involves much overlap between Machiavellianism and psychopaty. The main concern is that the short character inventory had only moderate internal consistency due to few items (n = 5 for each of 3 dimensions), but the size of the sample and the observed relationships indicate this was acceptable here.

Additional comments

The findings are well-presented and reasonable. Nicely written and an interesting contribution to abnormal and normal psychology.

---

## Round 0.2 · Minor Revisions

One of the 2 reviewers of your original submission has examined your resubmission; I too have reviewed this document. Both the reviewer and I agree that your manuscript has been improved as a result of your attention to the editorial team's comments. Pending your attention to a couple of minor 'basic reporting' issues noted by the reviewer, your manuscript will meet the criteria for publication in PeerJ. I look forward to receiving your revised manuscript in which you have addressed the minor reporting issues.

·

Basic reporting

The writing is generally clear. There are a few unclear sentences in the revised material. Using the tracked changes version,
In Results, page 9, after Table 1, the second sentence is incomplete (Despite the fact...). Should the period be a comma?
Table 1 and 2 in the files are the same. Check the numbering for correspondence with text after deleting one of the duplicates.
page 10, the sentence involving addition of "low tolerance towards others" is unclear as written. Should you add an "and" before "low tolerance..." or change Machiavellianism to an adjective?

Experimental design

Cross-sectional association study is appropriate for the goals.

Validity of the findings

The additional analyses confirm the core findings, which are crystal clear. It is very informative to have a comparison of light character profiles with the dark ones, which shows that the dark triad inadequately captures the anti-types already measured well by the light triad.

Additional comments

The revision was highly responsive and satisfies all reasonable criticisms. The limitations and strengths of the short character scale and the issues about using median splits were accurately described. I would have mentioned the concept of "attractor states" but that would have required more discussion of complex adaptive systems than may be justified here and you gave the appropriate references for this aspect of the discussion.

---

## Round 0.3 · accepted · Accept

Thank you for the prompt response and amendments based on the reviewer's comments. I'm now happy to proceed with publication of you manuscript. Well done.